# Prospective International Multicenter Pelvic Floor Study: Short-Term Follow-Up and Clinical Findings for Combined Pectopexy and Native Tissue Repair

**DOI:** 10.3390/jcm10020217

**Published:** 2021-01-09

**Authors:** Günter K. Noé, Sven Schiermeier, Thomas Papathemelis, Ulrich Fuellers, Alexander Khudyakov, Harald-Hans Altmann, Stefan Borowski, Pawel P. Morawski, Markus Gantert, Bart De Vree, Zbigniew Tkacz, Rodrigo Gil Ugarteburu, Michael Anapolski

**Affiliations:** 1Department of Obstetrics and Gynecology, University of Witten-Herdecke, Rheinlandclinics Dormagen, 41540 Dormagen, Germany; michael.anapolski@kkh-ne.de; 2Department of Obstetrics and Gynecology, University Witten-Herdecke, 258452 Witten, Germany; sven.schiermeier@uni-wh.de; 3Department of Obstetrics and Gynecology, St. Marien Hospital Amberg, 92224 Amberg, Germany; papathemelis.thomas@klinikum-amberg.de; 4Private Department of Surgical Gynecology, Krefeld (GTK) Germany, 47800 Krefeld, Germany; fuellers@gtk-krefeld.de (U.F.); Khudyakov@gtk-krefeld.de (A.K.); 5Department of Obstetrics and Gynecology, Regiomed Clinics Coburg, 96450 Coburg, Germany; harld-hans.altmann@klinikum-coburg.de; 6Department of Obstetrics and Gynecology, Clinic Links der Weser, 28277 Bremen, Germany; Stefan-Borowski@klinikum-bremen-ldw.de; 7Department of Obstetrics and Gynecology, Helios Clinic Bad Sarow, 15526 Bad Saarow, Germany; Pawel.Morawski@helios-gesundheit.de; 8Department of Obstetrics and Gynecology, St. Franziskus Hospital Ahlen, 59227 Ahlen, Germany; Markus.Gantert@gmail.com; 9Department of Obstetrics and Gynecology, ZNA Middelheim Antwerp, 2020 Antwerpen, Belgium; dr.devree@praktijkdevree.net; 10Department of Obstetrics and Gynecology, NHS Tayside Dundee, Dundee DD1 9SY, UK; ztkacz@nhs.net; 11Department of Obstetrics and Gynecology, University Hospital de Cabueñes, 33394 Gijon, Spain; guerillas3@hotmail.com

**Keywords:** prolapse, pelvic floor, laparoscopy, native tissue, pectopexy

## Abstract

Efforts to use traditional native tissue strategies and reduce the use of meshes have been made in several countries. Combining native tissue repair with sufficient mesh applied apical repair might provide a means of effective treatment. The study group did perform and publish a randomized trial focusing on the combination of traditional native tissue repair with pectopexy or sacrocolpopexy and observed no severe or hitherto unknown risks for patients (Noé G.K. J Endourol 2015;29(2):210–215). The short-term follow-up of this international multicenter study carried out now is presented in this article. Material and Methods: Eleven clinics and 13 surgeons in four European counties participated in the trial. In order to ensure a standardized approach and obtain comparable data, all surgeons were obliged to follow a standardized approach for pectopexy, focusing on the area of fixation and the use of a prefabricated mesh (PVDF PRP 3 × 15 Dynamesh). The mesh was solely used for apical repair. All other clinically relevant defects were treated with native tissue repair. Colposuspension or TVT were used for the treatment of incontinence. Data were collected independently for 14 months on a secured server; 501 surgeries were registered and evaluated. Two hundred and sixty-four patients out of 479 (55.1%) returned for the physical examination and interview after 12–18 months. Main Outcome and Results: The mean duration of follow-up was 15 months. The overall success of apical repair was rated positively by 96.9%, and the satisfaction score was rated positively by 95.5%. A positive general recommendation was expressed by 95.1% of patients. Pelvic pressure was reduced in 95.2%, pain in 98.0%, and urgency in 86.0% of patients. No major complications, mesh exposure, or mesh complication occurred during the follow-up period. Conclusion: In clinical routine, pectopexy and concomitant surgery, mainly using native tissue approaches, resulted in high satisfaction rates and favorable clinical findings. The procedure may also be recommended for use by general urogynecological practitioners with experience in laparoscopy.

## 1. Introduction

Due to controversies about the use of meshes, native tissue repair in pelvic surgery has currently rebecome the matter of choice in several countries. Native tissue repair was considered to be insufficient for a long period of time. However, several publications have shown that, from a clinical perspective, it provides better outcomes than meshes in the long term. In fact, the patients’ symptoms are improved to a much greater extent compared to the assessment of the sheer anatomical results [1,2,3]. Various vaginal or abdominal techniques (Manchester; sacrospinous fixation; high uterosacral fixation etc.) have been suggested for the restoration of apical support. To date, we lack validated data about the adequacy of these approaches. Sacral colpopexy with mesh is a frequently used technique in laparoscopy and has been evaluated in several studies. Due to the disadvantages of the approach (see below), our group devised the procedure of laparoscopic pectopexy in 2007 [4].

The so-called gold standard of laparoscopic sacral colpopexy (LSC) is based on several decades of extensive experience. The introduction of alloplastic material to fill the gap between the vagina and the sacrum accelerated the acceptance of the technique [5]. Extensive data have been reported from single-center studies, but a prospective multicenter trial comparing access and quality has not been published so far [6,7,8].

LSC commonly employs a y-shaped mesh deeply covering the total posterior length of the vagina and the anterior wall next to the bladder neck [9,10]. Comparison with published data is rendered difficult by the manifold approaches currently in use. Therefore, our group did focus on the use of mesh material only for apical support and did repair other defects with native tissue strategies [11].

Using pectopexy as apical support in combination with native tissue may reduce the risk of defecation disorders, which occur frequently after LSC. Additionally, mesh-related problems such as exposure at the vaginal wall were reduced [12].

De novo defecation disorders are anticipated in 17–34% of cases after LSC [9,13,14,15,16,17]. Slow intestinal transit, chronic flatulence, pain during defecation, and mild to severe constipation are the main symptoms reported in the literature. Published data on pectopexy have indicated the benefits of offering a standardized alternative option to LSC with the potential of reducing the risk of defecation disorders and bowel constriction by the mesh material, especially in obese patients [12]. The combination of native tissue repair and sufficient apical support leads to a low rate of de novo stress urinary incontinence (SUI) (4.5–7%) as well as minimal use of mesh material [3,12].

The present multicenter trial was performed to evaluate the effectiveness of the approach in general use by trained surgeons and determine the results of native tissue repair combined with apical mesh support in different hospitals and by different surgeons.

## 2. Materials and Methods

The study was initiated at 11 hospitals with 13 surgeons in four European counties. In order to ensure a standardized approach and obtain comparable data, all centers were instructed to use a prefabricated mesh (Dynamesh PRP 3 × 15) (approximately 25 cm²). In pectopexy, the mesh for apical support is fixed bilaterally at the pectineal ligament and anchored by sutures close to the crossing psoas muscle. This provides a fixation point at the level of the first sacral vertebra. Placement of the tape does not interfere with organs, vessels or nerves, and the defined fixation point ensures correct anatomical positioning of the vaginal axis. Owing to its position, the tape does not disturb the rectum or the hypogastric plexus. The diameter of the lower pelvis is not reduced by the technique.

Surgeons were trained by experts from the center at which the technique was developed, and data were collected on a secure server at the University of Wuerzburg. Every needed to have performed a minimum of 20 procedures before entering the study. All of the surgeons had private access to the server and could collect their data independently.

All patients who required surgical treatment (conservative treatment was either insufficient or was not accepted by the patient) were included in the study, except those with contraindications for laparoscopy. In accordance with common practice at the majority of the hospitals, the Baden-Walker classification (grades 1 to 4) was used to describe the defects. A distinction was made between apical defects, cystocele midline—cystocele lateral defects, and posterior defects. A modified version of the ICIQ-VS (International Consultation on Incontinence Questionnaire—Vaginal Symptoms) questionnaire was used to assess clinical complaints. The group did focus on complaints such as pelvic pressure, SUI, urgency, stool bulking/constipation, pain, and sexual impairment. Table 1 shows the rating of the complaints. Obstetric data and the patients’ histories of previous surgery, especially hysterectomy and cesarean section, were registered. Stress urinary incontinence (SUI) was stratified from grade 1 to 3, according to Stamey’s definition.

Concomitant surgeries, whether by the vaginal or laparoscopic approach, total operating times, and the time used for pectopexy, were registered. Intraoperative complications and postoperative data such as the duration of hospital stays, early—and late-onset infection (14 days after surgery), and wound infection were recorded in the database.

A total of 501 patients were registered in 14 months. Surgical data were analyzed and have been published recently [11]. Telephone interviews were not included in the evaluation. IBM SPSS statistics and Sigma plot Statistics (Systat Software, Inc., D-40699 Erkrath, Germany) were used for statistical evaluation.

## 3. Results

After the scheduled 14 months of data collection, 501 patients were registered on the server. Surgical and early complications have been reported in a previous publication [11]. Follow-up was performed 12–18 months after surgery (mean, 15 ± 2 months). A large number of patients, especially those at the main center, had to travel long distances (>100 km) or had difficulties arranging their transport, which had a negative impact on evaluation rates. Two hospitals, which had contributed two and 20 patients each, did not participate in the follow-up. More than 93% of the patients answered the follow-up inquiry. More than half of the patients who could not arrange to come answered the questionnaire, the others responded positively by phone. Only 7% did not react. Two hundred and sixty-four of 479 patients (55.1%) underwent a physical examination and were followed up with the questionnaire used prior to surgery. A distinction was made between de novo and persistent complaints and defects.

The distribution of concomitant surgeries in the examined group was similar to that in the entire study group. Table 2 shows the total and relative numbers for all surgical approaches in the entire group compared to the examined group. The distribution of defects was similar in both groups (Table 3). Table 4 shows primary symptoms preoperatively and during follow-up.

### 3.1. Pelvic Pressure

Only 4.1% of patients with a preoperative sensation of pelvic pressure reported no significant change after surgery (chi-square *p* < 0.001), while 5.7% reported de novo pressure due to relapse or de novo changes. Previous symptoms were reduced in 95.9% of patients.

### 3.2. Urgency

50% of the patients included in the study reported urgency before surgery, whereas 86% of this group had no urgency after surgery (chi-square *p* < 0.001). De novo urgency was registered in 4.2% of patients. 25% (34) had an additional loss of urine prior to surgery. One (9%) of the patients with de novo urgency also had a first-degree loss of urine. In the group with urgency persistence, 3 (16%) women with persistent grade 1 incontinence were registered.

### 3.3. Sexual Impairment

Sexual impairment was considered relevant by 16.8% of the total cohort (15.9% of the follow-up group). Only 7.14% of patients complained of persistence after surgery.

### 3.4. SUI

Twenty-four percent of the patients reported SUI before surgery. SUI was rated: grade 1 by 25%, grade 2 by 66%, and grade 3 by 9% of patients. Of those who underwent additional surgery (colposuspension *n* = 43), 72.1% of patients were dry, and 93% were improved; 7% reported persistence of previous symptoms after surgery. Thirteen patients did not receive any treatment for incontinence; 53.4% were dry after prolapse surgery; and 30.8% reported persistence of their previous incontinence. Only two cases (0.8%) of de-novo incontinence were identified in the whole cohort.

### 3.5. Stool Bulking and Constipation

Preoperatively, 11.2% of patients experienced stool bulking or constipation. We noted persistence in 25.0%, and one patient with de novo symptoms.

Since a slight degree of under-correction of level I. (between grade 0–1) was agreed upon in order to avoid the side effects of over-correction, grade 1 was considered to indicate cure of presurgical stages 2 and above. Cure was registered in 94.3% of patients, while 96.6% were either cured or improved.

### 3.6. Level II

Out of 35 untreated grade 1 cystoceles, 15 (42%) persisted while two (6%) deteriorated (Grad 2). 133 patients with a grade 2 or higher midline defect received an additional native tissue repair. 121 (91%) showed cure or improvement, while 12 persisted or worsened.

99 patients with clinically relevant posterior defects were treated with additional native tissue repair. In this group, 94 (95%) showed an improvement or cure.

When asked “Would you recommend the treatment to a relative?” the question was answered positively by 95.1% of patients. The mean rating on an analogue satisfaction scale from 1–10 was 8.7. Overall, 90.2% of patients gave a rating between 7 and 10. The reasons for not recommending the treatment were pelvic pressure (persistent or de novo) (eight cases), de novo pain (two cases), and de novo sexual impairment (three cases, two of which involved persistent incontinence).

### 3.7. Complications

Three lymphatic seromas at the lateral suspension site were treated by laparoscopy. One TVT was placed after 7 months because of incontinence. Two re-interventions were performed by the laparoscopic approach because of early level 1 recurrence. One patient with urinary retention received medical treatment. One de novo enterocele and one de novo cystocele were operated on during the follow-up period. No mesh exposure or mesh complication was observed during follow-up.

## 4. Discussion

### 4.1. Main Findings

Previous publications have indicated that pectopexy is safe and can be incorporated in clinical routine [11,12]. Since controversies concerning meshes may also affect abdominal techniques with the extended use of meshes (deep anterior or posterior mesh placement), Our group collected data on reduced mesh use adding to the effectiveness of pectopexy. In this study, 15 cm of PVDF tape (Dynamesh PRP 3 × 15) were used solely for apical support (approximately 25 cm²). In a computer simulation, Bhattarai et al. showed that bilateral fixation in pectopexy permits better physiological positioning of the bladder and vaginal cuff than unilateral sacral colpopexy during the Valsalva maneuver [18].

The anchor point of pectopexy lies 1–2 cm above the natural apex and does not allow for correction of a cystocele or a posterior defect by pulling the vagina cranially. Therefore, in this study, our group did combine apical support with concomitant repair, depending on individual defects and disorders (Table 2). Notably, no additional mesh was used for cystocele, rectocele, or enterocele repair.

The results were compared with those of LSC, mainly performed with deep mesh fixation [9,10,19]. The comparison was rendered difficult by inconsistent results and the absence of prospective multicenter trials. Therefore, the group did compare its findings mainly with reports from single-center studies, which comprised small sample sizes and patients who did not undergo physical examination. The majority of published studies have been focused on anatomical changes or outcomes; clinical findings were given less importance.

The cure rate in this study for level 1 was 94.3%, while 96.6% of patients were either cured or improved. This is a confirmation of previous data [12]. Similar rates have been reported for LSC (94–100%) [10,20,21,22].

The primary symptom was pelvic pressure and bulging, which occurred in 86.7% (*n* = 229). Persistence of this symptom was noted in 4.1% of patients. Symptoms were reduced in 95.2% (*p* ≤ 0.001 for the chi square test). De novo symptoms, mainly due to de novo pelvic floor defects, were reported by 5.7% of patients. Liedel et al., who studied 277 patients, noted reduced symptoms in 82.7% (*p* = 0.00001) after vaginal pelvic floor surgery [23]. In retrospective data, Bojahr et al. noted a reduction of symptoms in 90.7% of patients after LSC [24].

This study did register a reduction of pain in 98% of patients, whereas Liedel et al. observed the same in 53.1%, and Bojahr et al. in 44% of patients. These diverse outcomes may have been due to different interpretations of the sensation of pain.

Of the follow-up group, 51.7% (136) of patients complained of urgency and frequency before surgery. This symptom was reduced in 86% of patients during follow-up; 4.2% complained of de novo urgency. Several studies have focused on urgency and prolapse. Whilst Malanowska et al. reported a reduction of symptoms in 76% and a de novo rate of 2.6% of patients for the lateral suspension technique, Illiano noted a reduction of symptoms in 73.6% of patients after LSC and Rexhepi et al. observed a reduction in 67% of patients who were treated with bilateral LSC. Compared to published success rates for pectopexy, we achieved excellent results [25,26,27].

The measurement of sexual impairment was one of the most problematic issues. Pelvic floor defects are associated with a combination of anatomical obstacles and psychological embarrassment. Prolapse problems could not be distinguished easily from interpersonal incompatibility in sexual relationships. The data are quite heterogeneous. Our group did observe different outcomes in its studies; 15.9% of the follow-up cohort complained of sexual impairment before surgery. The reduction registered in 92.9% of patients was surprisingly high, but a de novo problem occurred in 2.7%. Half of the latter patients would not recommend the procedure because of their de novo sexual impairment.

This study measured a reduction of stool bulking and constipation in 75% of patients (*p* < 0.001). Whilst studies addressing vaginal repair have reported a reduction of symptoms in 66% of patients after vaginal repair, most LSC studies mention an increased rate during follow-up [16,17,23,24,28,29]. As LSC is known to be associated with bowel symptoms, vaginal repair, as well as pectopexy, appear to significantly improve this complaint. The results support previous findings from our first randomized trial [12].

De novo incontinence occurred in 0.8% of cases, and 56 patients had relevant incontinence before surgery. A total of 43 patients were treated simultaneously with colposuspension, whereas 13 did not receive additional treatment. Improvement was registered in 93% of the first group, but only 53.4% of the second group (*p* = 0.003; Fisher’s exact test). High incontinence rates (5–40%) have been reported after LSC. Some authors recommend adding Burch colposuspension to the surgical strategy. However, this topic remains controversial [20,24,30,31,32]. Our findings support simultaneous treatment in a multiple compartment setting.

There are some long-term studies on sacropexy or hysterosacropexy available which report a high level of safety in the procedures. The latter technique enables an exposure rate of only 0.4% with a median observation of 46 months (multicenter questionnaire study). Nightingale and Phillips examined 93 of 112 patients over 9 years of age (mean 6 years follow-up) and reported only one mesh complication (bowel obstruction). We did not find any mesh complication yet and hope to publish a long term follow up soon [33,34].

### 4.2. Strengths and Limitations

The multicenter setting provided a large cohort of patients who could be studied prospectively at a large number of international centers. Clinically relevant anatomical, functional, and subjective findings are reported here. Due to the heterogeneity of clinical practices in four countries and 11 centers, and to ensure the comparability of data, we used a standardized questionnaire designed for the study in addition to routine data collection. This limits the comparability of the present investigation with other publications based on questionnaires provided by the International Urogynecology Association (IUGA) or International Continence Society (ICS). An international trial entails the inclusion of different experiences and diverse traditions. We could standardize the technique for pectopexy and laparoscopic cystocele and rectocele repair, but the vaginal approaches were based on local experience. The effect of these differences on the collected data could not be measured.

### 4.3. Interpretation

A positive recommendation rate of 95.1% and a mean satisfaction rate of 8.7 (from 1 to 10) expressed the high degree of clinical acceptance by the patients. Figure 1 shows the distribution of the rating scale for the entire treatment. The negative recommendations resulted from de novo sexual impairment (3), de novo defects, and relapse. Complications such as infections or seroma were widely accepted.

## 5. Conclusions

Pectopexy combining apical support with a prefabricated PVDF tape and native tissue repair for level 2 and 3 defects yielded favorable clinical outcomes and a low re-intervention rate after a mean follow-up period of 15 months. A prospective international multicenter study provides valid results because of the large sample size and the standardized procedures performed by independent surgeons. Given the favorable results and the low rates of side effects, the approach may be recommended as an alternative to LSC for experienced surgeons. It provides the option of reducing mesh use by combining adequate apical support with native tissue repair. The long-term follow-up should permit the identification of those patients who require additional mesh for level 2 and 3 treatment.

## Figures and Tables

**Figure 1 jcm-10-00217-f001:**
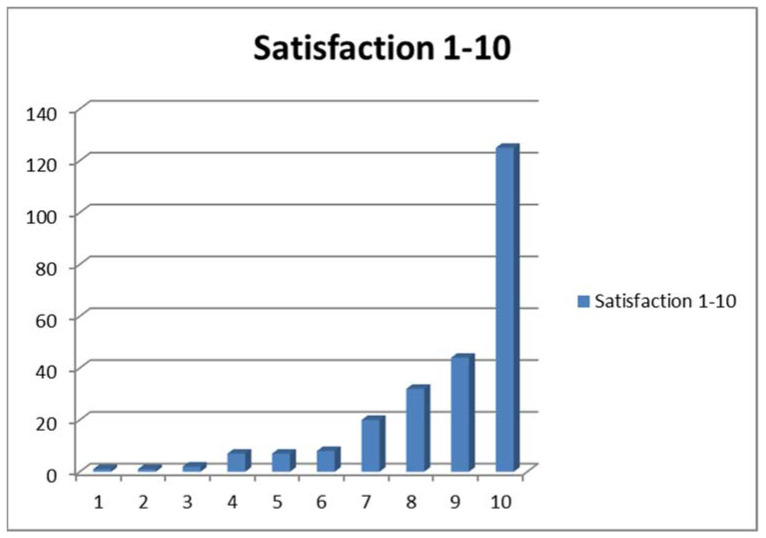
Rating scale for patients’ satisfaction with their treatment.

**Table 1 jcm-10-00217-t001:** Measurement of symptoms by the questionnaire based on International Consultation on Incontinence Questionnaire—Vaginal Symptoms (ICIQ-VS).

	Measurement	
Complaints	Positive	Negative
Pelvic pressure	Daily or regularly	Rare or no pressure
SUI	Stratification by Steamy	No SUI
Urgency	Positive answer and bothersome for the patient	No nycturia or frequency, no urge feeling
Stool bulking	Feeling of pressure in the lower rectum	No rectal problems
Constipation	Need for laxatives, slow transit	Normal defecation
Pain	Pain in the pelvis	No pain

**Table 2 jcm-10-00217-t002:** Concomitant surgeries in the entire trial group and the physical examined follow-up group.

Study Group Total/Follow-Up Approach	Frequency Entire Group	Percentage	Frequency in Follow-Up (Examined) Group	Percentage
Pectopexy	501	100	264	100
Laparoscopic cystocele repair	173	34.5	83	31.5
Laparoscopic posterior repair	132	26.3	58	22.0
Vaginal anterior repair	68	13.6	50	18.9
Vaginal posterior repair	59	11.8	41	15.5
Laparoscopic lateral repair	115	22,9	50	18,9
Burch colposuspension	64	12.8	34	12.9
Vaginal tape	2	0.4	2	0.76
LSH	313	62.5	175	66.3
TLH	5	1.0	2	0.76

**Table 3 jcm-10-00217-t003:** Distribution of defects in the total study group compared to the examination group.

Defect	Total Group	Examination Group
Apex grade 2	57%	52%
Apex grade 3 and 4	37%	39%
Cystocele grade 2 and 3	60%	59%
Posterior grade 2	12%	11%
Posterior grade 3	13%	14%

**Table 4 jcm-10-00217-t004:** Impact of surgery on complaints in the follow-up group before and after surgery.

	Pre-Surgery	Follow-Up	De Novo	Persistent
Pelvic pressure	86.7% (229)	9.8% (26)	5.7% (15)	4.1% (11)
Pain	18.9% (50)	2.7% (7)	2.3% (6)	0.4% (1)
Urgency	51.7% (136)	11.4% (30)	4.2% (11)	14% (19)
Sexual impairment	15.9% (42)	3.4% (9)	2.7% (6)	7.1% (3)
Stool bulking/	11.2% (28)	3.2% (8)	0.4% (1)	25.0% (7)

## Data Availability

Data available on request due to restrictions e.g., privacy or ethical.

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
