# Peer review of "Prospective International Multicenter Pelvic Floor Study: Short-Term Follow-Up and Clinical Findings for Combined Pectopexy and Native Tissue Repair"

_jcm, 2021, doi:10.3390/jcm10020217_

Round 1

Reviewer 1 Report

This is a well written multi-centre observational cohort study involving 11 centres, 4 countries and 501 patients, of which 55% were available for examination at 12-18 month follow up.  The technique for pectopexy and native tissue repair is relatively new and novel and has recently been described and published (ref 11).  The paper is written in excellent English and has a sound comprehensive hypothesis with logical introduction and methodology. The results on the whole are well presented and the discussion addresses salient controversies in the literature as well as strengths and weaknesses in the study.

As this is newer technique, publications outlining quality data on efficacy and safety are of high importance.

I have a few minor comments / questions:

The centres utilised Baden Walker rather than PoPQ classification for prolapse which is a recognised classification although less standardised.  The authors report symptom outcomes but no clinical examination data from the 55% that attended for follow up

This would be useful and should be included, along with recognition that the BW classification is probably more pragmatic in multicentre studies (Eg VUE or PROSPECT studies) but less standardised.

Were there any exposures of the mesh in those patients attended for follow up?

Table 2 is difficult to understand and needs clarification.  I can not understand what the "frequency Follow up" column relates to.  Please can this be clarified better in the paragraph 137-141.

The discussion discusses comparisons on the described technique with laparoscopic sacrocolpopexy, however would benefit from mentioning safety and importance of long term follow data.  Pain score are very low in the cohort which is important in the current climate and scrutiny of mesh procedures.  The technique described does involve a mesh (albeit smaller volume and only fixed to the apex).  Acknowledgement of long term data on success and safety is important (some of the references are quite old)

Recent 2020 publications from Izett et al, and Nightingale et al both report upto 10 year  outcome data.

(Izett-Kay ML, Aldabeeb D, Kupelian AS, Cartwright R, Cutner AS, Jackson S, Price N, Vashisht A. Long-term mesh complications and reoperation after laparoscopic mesh sacrohysteropexy: a cross-sectional study. International urogynecology journal. 2020 Dec;31(12):2595-602.) 

(Nightingale G, Phillips C. Long-term safety and efficacy of laparoscopically placed mesh for apical prolapse. International Urogynecology Journal. 2020 Jun 10:1-7.)

On the whole this is a valuable piece of work and warrants publication pending a few minor amendments.

Author Response

Attached all my answers to your Questions and the changed manuscript.

Reviewer 2 Report

This is an interesting article on clinical findings after combined pectopexy and native tissue repair (multicentre pelvic floor study).

I have some comments:

Did you include patients with previous pelvic surgery? If so, did these patients demonstrate any difference in post-op results compared to those without?

Tables 2 and 3

Table 3 shows that 60% of the entire trial group had a pre-op grade 2 or 3 cystocele. But table 2 shows that 48.1% of the sample underwent laparoscopic or vaginal anterior repair. Please explain this apparent discrepancy.

Table 4

Table 4 shows that 29 patients had pelvic pressure postoperatively: in 11 cases they had persistent pelvic pressure and in 15 cases de novo. Please check the numbers (15 + 11 equals 26).

Page 5

Were there any cases of UUI preoperatively?  If no specify this in the text. If yes specify the incidence of this symptom postoperatively.

Finally, I think it would add value to your work if you specified in detail the anatomical results of your surgery.

Author Response

Attached the answers to the questions and the revised manuscript
